# Newly Developed Restorer Lines of Sorghum [*Sorghum bicolor* (L.) Moench] Resistant to Greenbug

**DOI:** 10.3390/plants13030425

**Published:** 2024-01-31

**Authors:** Evgeny E. Radchenko, Irina N. Anisimova, Maria K. Ryazanova, Ilya A. Kibkalo, Natalia V. Alpatieva

**Affiliations:** N.I. Vavilov All-Russian Institute of Plant Genetic Resources, 190000 St. Petersburg, Russia; eugene_radchenko@rambler.ru (E.E.R.); ryazanovamk@yandex.ru (M.K.R.); i.kibkalo@vir.nw.ru (I.A.K.); alpatievanatalia@mail.ru (N.V.A.)

**Keywords:** grain sorghum, CMS A1 (milo), *Rf* genes, molecular markers, breeding, valuable traits, *Schizaphis graminum*

## Abstract

Eight lines of grain sorghum [*Sorghum bicolor* (L.) Moench], which can be used as a promising source material in heterotic hybrid breeding as pollen fertility restorers and donors of resistance to the greenbug (*Schizaphis graminum* Rondani), are characterized. The new restorer lines (R-lines) were developed by crossing the maternal sterile line Nizkorosloe 81s (CMS A1) with two lines selected from the grain sorghum collection accessions VIR-928 and VIR-929 as the paternal forms. The R-lines were genotyped using PCR markers, and also characterized by height, duration of the seedling–flowering period, and some of the technological properties of flour. With the use of microsatellite markers linked to the *Rf* genes and by hybridological analysis, it was shown that the new lines carry the dominant allele of the gene *Rf2*. The PCoA analysis demonstrated clear differences of each R-line from the parents. The genotypes of the new lines and their parental forms for the *Rf2* locus were confirmed by applying three allele-specific codominant CAPS markers which detected SNPs in the candidate *Rf2* gene. All new lines were highly fertile, as demonstrated by cytological analysis of acetocarmine-stained pollen preparations. A high resistance to the greenbug was demonstrated for each new R-line both in the laboratory and field conditions against a severe aphid infestation. Grain quality parameters such as protein content and dough rheological properties varied widely and were quite satisfactory in some R-lines. Characteristics common to all eight sorghum lines studied, such as the ability to restore pollen fertility in the F_1_ generation, good pollen quality, greenbug resistance, early ripening, spreading panicle, and low stature, allow us to recommend them for producing commercial F_1_ hybrids with satisfactory grain quality for both fodder and food purposes.

## 1. Introduction

Sorghum [*Sorghum bicolor* (L.) Moench] grain is a staple food for millions of people in African and Asian countries. Owing to its high heat tolerance and drought resistance, sorghum is successively cultivated in regions with arid and semi-arid climates. Recently, due to global climate change, accompanied by frequent droughts, sorghum is increasingly considered a valuable insurance crop. In Russia, as in European countries, sorghum is cultivated mainly as forage crop. Moreover, in the Russian food industry, sorghum grain is used as a wheat substitute for the production of gluten-free products. Recently, however, interest in sorghum has increased due to its potential use as a source of slow-digesting starch in dietary nutrition. In Russia, many varieties of sorghum were created by selection from hybrid populations from intervarietal and interspecific crosses.

The creation of hybrids is one of the promising areas of modern sorghum breeding. In the 1950s, after the discovery of cytoplasmic male sterility (CMS) in sorghum [1], F_1_ hybrids became widespread in commercial seed production. The yield of such hybrids exceeded the yield of conventional varieties by an average of 20–40%. In sorghum heterotic hybrid breeding, the CMS A1 (milo) being predominantly used is the result of interactions between the durra grain sorghum cytoplasm and nuclear genes of the kaffir race [2]. For producing F_1_ hybrid seeds, sterile CMS lines are crossed as maternal forms with paternal lines to create fertility restorers. Genes for pollen fertility restoration (*Rf*), introgressed to the hybrid genotype from paternal restorer lines, block or compensate for the effects of specific mitochondrial products, and probably ensure coordinated activity of the nucleus and cytoplasm [3]. Alternative cytoplasm types, for example, A2, A3, 9E, and others, are not yet widely utilized in commercial seed production due to the limited number of effective restorer lines and significant variability in fertility restoration depending on environmental conditions [4].

Until 1960, Russian breeders had only one sterile line A-385 (USA) based on CMS A1 (milo) at their disposal; then, by 1976, 95 sterile analogues had already been developed in domestic and foreign breeding centers, and their number has increased every year. The widespread use of sterile lines with the same cytoplasm type (A1) and utilizing a narrow set of paternal lines/fertility restorers result in a reduction in the genetic diversity of sorghum cultivated hybrids and consequently in a decrease in their resistance to diseases and pests. Exploiting new sources of pollen fertility restoration genes for CMS A1 will undoubtedly broaden the genetic diversity of hybrids.

In sorghum, several genes localized on different chromosomes have been identified that restore pollen fertility in A1 CMS-based hybrids. The *Rf1* locus was mapped to chromosome SBI-08 [5,6], *Rf2* to SBI-02 [7], *Rf5* to SBI-05 [8], and *Rf6* to SBI-04 [9]. The most plausible candidates within each *Rf* locus encode proteins of the PPR (pentatricopeptide repeat) family containing repeated motifs of 35 amino acids, whose functioning is necessary for the restoration of the pollen fertility of F_1_ hybrid plants. In order to be in demand for breeders, such lines, in addition to the presence of *Rf* genes in their genotypes, must correspond to the CMS lines in terms of height and duration of the seedling–flowering period, and also possess economically valuable traits, such as resistance to harmful organisms, and good grain quality indicators.

To assess a potential paternal line for the ability to restore male fertility, it is crossed with a sterile maternal line and the pollen fertility of F_1_ plants is estimated. It is known that in most cases, the pollen fertility restoration trait is controlled by dominant alleles of the *Rf* genes. If the paternal line has the *Rf* dominant allele, the resulting F_1_ plants will produce pollen; if the paternal line possesses the recessive allele (in sterility maintainers), then the F_1_ plants will also be sterile. The process of developing fertility restorers and sterility maintainers (carrying dominant or recessive alleles of *Rf* genes, respectively) can be accelerated by using molecular markers. Markers used for screening breeding material for the presence of *Rf* genes must be allele-specific, codominant, and characterized by a good reproducibility and insensitivity to DNA quality. These requirements are met by a number of PCR-based markers, the most efficient of which are SSR (Single-Sequence Repeat) markers linked to the target genes and SNP (Single-Nucleotide Polymorphism) markers developed for detecting differences in alleles using single polymorphic nucleotides [10].

It is known that sequences of *Rf* genes are clustered on the chromosomes and highly polymorphic. The allelic variants of candidate genes in the *Rf* loci differ by multiple SNPs. To utilize these SNPs for marker-assisted selection, KASP (competitive allele-specific PCR) or the more accessible CAPS (cleaved amplified PCR sequences, or PCR-RFLP) markers are being elaborated [11]. In various crops, including sorghum, SSR markers linked to the *Rf* genes have also been developed and effectively used [5,6,7,8,9,12].

In the collection of the N.I. Vavilov All-Russian Institute of Plant Genetic Resources (VIR), tall, late-maturing grain sorghum accessions from China, VIR-928 and VIR-929, were found, which each have two highly effective dominant genes for resistance to a key pest of sorghum, the greenbug *Schizaphis graminum* Rondani. VIR-928 also possesses a third dominant gene that is expressed against individual aphid clones [13]. R-lines 928-1 and 929-3 were selected from these accessions. They are characterized by aphid resistance and the ability to restore the pollen fertility of F_1_ hybrids in crosses with A1 type-CMS lines. Subsequently, based on hybrids from crossing these lines as paternal parents with the CMS line Nizkorosloe 81s, new early-maturing R-lines characterized by optimal height and resistance to the aphid were derived. The work lasted 10 years. For the further use of these lines as donors of valuable agronomic traits in breeding programs or as pollinators for F_1_ hybrid seed production, it was necessary to confirm their genetic homogeneity, to identify genotypes with the major loci associated with the CMS-*Rf* genetic system, to phenotype in terms of the plant architecture-related traits and duration of seedling–flowering period, and to characterize their seed quality.

## 2. Results

### 2.1. Phenotypic Characterization of New Sorghum Lines

In sorghum heterotic hybrid breeding, R-lines are in demand, showing a flowering time that is the same or several days in advance compared to sterile lines. The similarity of the parental lines in terms of their height is also relevant for better pollination; the height of the R-lines should be identical to or slightly exceed that of the maternal lines. In our study, three short-height sterile lines were used as controls when both measuring plant height and determining the germination–emergence period of the new R-sorghum lines Nizkorosloe 81s, A-83, and A-10598. For better pollination, a preference is given to R-lines with a spreading panicle and long pollen filaments. Figure 1 shows the results of the phenotyping of eight new lines and their parental forms in terms of plant architecture traits, the height of the main stem, and the panicle shape. Based on the results of the two-year assessments, the length of the main stem of the new R-lines was significantly shorter than that of the original fertility restorer lines 928-1 and 929-3 (95–125 cm in new lines compared to 260–310 cm in the original genotypes). The value of this trait is optimal for successful pollination of the potential maternal sterile lines Nizkorosloe 81s, A-83, and A-10598, whose height is generally in the range of 100–108 cm (Figure 1C,D).

The new restorer lines and the original accessions differ by panicle shape. Unlike the dense “lumpy” panicle in VIR-928 and VIR-929, the new lines are characterized by a loose pyramidal panicle with long pollen filaments that make them more suitable for the pollination of the male sterile lines Nizkorosloe 81s, A-83, and A-10598 possessing the A1 (milo) cytoplasm. In 2021 and 2022, the measured panicle length and width varied widely, so we used the panicle aspect ratio (length-to-width ratio) for comparison. According to the results of the two-year measurements in triplicate, the ratio of panicle length to width of the original accessions VIR-928 and VIR-929 ranged between 1.4 and 2.3, while the panicle aspect ratio of the newly developed lines ranged between 3.43 and 7.5 (Figure 1A,B).

In the field experiments performed in 2022 at KES VIR, all new lines showed early flowering. The R-928-3 line manifested the shortest seedling–flowering period (58 days). The lines R-929-1, R-929-2, R-929-3, R-928-1, R-928-2, R-928-4, and R-928-5 initiated flowering a little later, after 62 d of vegetation, whereas the original restorer lines 929-1 and 928-3 flowered 73 d. after seedling emergence. Thus, the 58–62 d flowering time of the newly developed lines was optimal for pollinating the sterile lines Nizkorosloe 81s, A-83, and A-10598, which flowered in 56 d.

A high level of male fertility is the most important trait for pollinator lines used in the production of commercial hybrids. To assess the fertilization potential of the newly developed lines, the pollen preparations were stained with acetocarmine. The percentage of stained pollen grains (pollen fertility) was high in each of the studied lines and exceeded 90% (Figure 2).

### 2.2. Phenotypic Evaluation for Aphid Resistance

All adult plants of the studied lines showed high resistance to *S. graminum*, comparable to that of the paternal lines 928-1 and 929-3, both with a strong aphid infestation of plants in the field and under laboratory experiments when assessing juvenile resistance of the seedlings. In the field, the average damage to 10 randomly selected plants did not exceed three points, while in the Nizkorosloe 81s line it turned out to be significant and was estimated at five points (Figure 3A). In the laboratory conditions, when Nizkorosloe 81s plants died, the damage to the leaf blades of young plants of the new lines did not exceed 10% and were estimated at 0, 1, or 2 points (Figure 3B).

### 2.3. Genotyping of Sorghum Lines

To confirm the genetic diversity of the new R-lines, seven microsatellites and two CAPS markers that were polymorphic between the original parental forms were used for genotyping (Appendix A). Three to five plants were analyzed for each sample. No differences were found among the plants within the same sample, indicating the homozygosity of each studied line. The identical marker profiles were obtained for the original paternal lines 928-1 and 929-3, for the lines R-929-1 and R-929-2, and for R-928-1 and R-928-2, as well as for R-928-3, R-928-4, and R-928-5. The R-929-3 line had a unique profile. Thus, all R-lines differed from the parental lines 928-1, 929-3, and Nizkorosloe 81s. A PCoA plot was generated using GenALEx 6.5 based on the results of molecular genetic analysis; a grouping of the studied genotypes is shown in Figure 4.

### 2.4. Identification of Alleles of the Rf Genes in the Original Sorghum Genotypes 928-1 and 929-3, and in New Lines

For the effective use of new R-lines in breeding programs, codominant molecular markers are needed to identify pollen fertility restoration genes and control their presence in the source breeding material, and to differentiate plants homozygous and heterozygous for the *Rf* loci in the hybrid populations.

In samples of the sorghum original parental forms, eight highly polymorphic microsatellite markers linked to the *Rf* genes were amplified: Xtxp18, Xtxp250 (*Rf1*), Xtxp50, MS-37912, MS-Sb3466, MS-Sb 3460 (*Rf2*), Xnhsbm1089, (*Rf5*), and SB2387 (*Rf6*) [5,6,7,8,9,14]. Four markers (Xtxp18, Xtxp50, Xnhsbm1089, SB2387) turned out to be polymorphic between the sterile line Nizkorosloe 81s and the original restorer lines 928-1 and 929-3. These SSR markers were selected for the analysis of the F_2_ population from crossing the Nizkorosloe 81s line and the fertile line 929-3 (Appendix A). Each of the 80 F_2_ plants was phenotyped for fertility/sterility traits and further genotyped using a series of markers. Good-quality profiles of the markers Xtxp18, Xtxp50, Xnhsbm1089, and SB2387 were obtained for 80, 80, 66, and 53 plants, respectively.

To determine the genotypes of the parental forms by the pollen fertility restoration loci, we verified a working hypothesis about the presence of dominant alleles of the *Rf* gene (genes) in line 929-3, whose activity in hybrids with the A1 cytoplasm is necessary for pollen fertility restoration and, as a result, for setting seeds in isolated inflorescences. On the contrary, the sterile maternal line Nizkorosloe 81s possesses recessive alleles of the *Rf* genes. If the marker and the fertility restoration trait are inherited independently, four phenotypic classes were expected to appear (male fertile plant/paternal variant of the marker; male sterile plant/maternal variant of the marker; male sterile plant/paternal variant of the marker; and male sterile plant/maternal variant of the marker) in the ratio 9:3:3:1. The results of the analysis of the correspondence of the observed segregations in the F_2_ to the theoretically expected ones using the chi-square test confirmed the independent inheritance of the pollen fertility restoration trait and the markers Xtxp18 (*Rf1*), Xnhsbm1089 (*Rf5*), and SB2386 (*Rf6*). At the same time, a pollen fertility restoration trait was inherited that was linked to the Xtxp50 marker of the *Rf2* gene (χ^2^ = 48.65 *p* < 0.005) (Table 1). It can be assumed that it is the *Rf2* gene that determines the ability to restore pollen fertility in the parental line 929-3. The SSR marker Xtxp50 of the *Rf2* dominant allele has a length of 317 bp, and the marker for the recessive *rf2* allele has a length of 304 bp (Appendix A). The exceptions were four plants: three of them with the maternal type of the Xtxp50 marker turned out to be fertile, and one plant with the paternal type was sterile, which can be explained by possible recombination between the *Rf2* gene and the marker allele.

### 2.5. Development of CAPS Markers and Rf2 Alleles Confirmation

In the protein-coding sequence of the *Rf2* candidate gene Sobic.002G057050 from parental lines 928-3 and 929-1, several SNPs were found, including substitutions of adenine for thymine at the positions 1090 and 1091 [11,15], which made it possible to develop two intragenic CAPS markers to identify alleles of the *Rf2* gene in the material under study. The sterile line Nizkorosloe 81s has sites of recognition by the restriction enzymes MseI and HpaI in the interval between the nucleotides 1087 and 1097 of the Sobic.002G057050 sequence, whereas the restorer lines 928-1 and 929-3 lack the sites in that sequence fragment (Appendix A). The fragments obtained after digestion by the restrictase MseI of the PCR product amplified using primers 2459403fw and 2459403rev (see Section 4.4.2) were different for sterile and restorer lines. The +572 bp restriction fragment is characteristic of the dominant allele (*Rf2*), but it was absent from the fragments obtained after restrictase treatment of the amplification product of the recessive (*rf2*) allele. After the treatment of the same amplicons by the restriction enzyme HpaI, a single fragment of 935 bp length identified the *Rf2* allele, and two fragments of 753 + 182 bp lengths were indicative of the *rf2* allele. However, due to the fact that primer annealing sites were present not only in the Sobic.002G057050 sequence, but also in the homologous sequences that lacked HpaI restriction sites in the same nucleotide interval, the restriction products of the sterile line amplicon contained three fragments: 935 + 753 + 182 bp. Thus, only when the two markers were used simultaneously was it possible to identify a *Rf2/rf2* heterozygote in a segregating population (Figure 5).

Both SSR and CAPS markers were equally effective for identifying alleles of the *Rf2* gene. As expected, the sterile plants of the F_2_ population (Nizkorosloe 81s × 929-3), with one exception, had maternal variants of the SSR marker Xtxp50 and both CAPS markers. The marker profiles of all fertile plants, the putative heterozygotes (genotype *Rf2rf2*), included fragments from both parents. Fertile, presumably homozygous *Rf2Rf2* plants were characterized by the presence of paternal-type markers (Table 2).

### 2.6. Analysis of Baking Properties of Sorghum Flour

In order to characterize the new sorghum lines as potential sources of meal for feed and the food industry, each sample was assessed according to the following flour traits: protein content and stability of the protein complex (the sedimentation test). In addition, using a farinograph, the rheological properties of dough obtained from the composite flour with wheat were assessed.

The protein content of the lines varied slightly and averaged 11.54 ± 1.04%. At the same time, the quality of sorghum flour differed among the genotypes. Based on the results of sedimentation analysis, the volume of sediment of the first phase of the test in acetic acid was 38.6 mL, with minor variations between samples and replicates. However, after adding SDS, only the R-928-1 line turned out to be resistant to such an increase in physicochemical load, while the other lines, on the contrary, increased the volume of sediment and the SDS/AA ratio varied from 1.2 to 1.3 (Appendix A), i.e., according to the type of swelling of the ground grain, they belonged to the third class of grain sorghum [16].

To determine the prospects for using new lines in breeding programs for food production as improvers of the baking properties of wheat flour, we compared the rheological properties of dough from wheat flour containing 10 or 30% of sorghum flour: dough formation time, dough stability, dough liquefaction, the farinograph quality number, and valorimeter value. The quality parameters significantly increased for wheat samples with sorghum flour added in comparison with the control. On the contrary, the dough liquefaction values decreased significantly, which also indicates an improvement in the dough quality, since these characteristics have a negative expression (that is, the lower the indicator, the better the quality). Appendix A presents the average values of the rheological characteristics of the composite flour samples. Rheological properties of the composite flour containing 10% of flour from sorghum lines varied insignificantly, but after adding 30% sorghum flour, significant differences were revealed. Thus, the R-928-1 line showed the worst quality indices, and the line R-928-4 had a farinograph quality number significantly higher than the average for the sample, as well as the highest valorimeter number in the sample and the lowest dough liquefaction parameter.

## 3. Discussion

The new greenbug-resistant sorghum R-lines, derived from crossings of the sterile line Nizkorosloe 81s (CMS A1) with lines selected from the grain sorghum accessions VIR-928 and VIR-929 (Dzhugara white, Western China) from the VIR collection, are homozygous, characterized by genetic diversity and have a number of valuable breeding traits. With the use of molecular genetic markers and hybridological analysis, the presence of the *Rf2* fertility restoration gene in their genotypes was confirmed. These lines possess valuable agronomic traits and therefore can be utilized directly for producing hybrids or as sources of the *Rf2* gene in breeding new restorers. The allele-specific molecular markers developed based on the SNPs in the candidate gene sequences, as well as the SSR markers closely linked to the *Rf2* gene, can be used for controlling F_1_ seed hybridity, determining the purity of R-lines, and for the analysis of populations and isolating homozygous plants from them. There are numerous examples of developing the diagnostic molecular markers for fertility restorer genes in sunflower [17], pepper [18], onion [19], rapeseed [20], and other crops [21]. Currently, several SSR markers have been developed which are intensively used for the identification of the *Rf2* gene in sorghum collection and breeding material [22,23]. The coding sequences of the candidate gene Sobic.002G057050 in the *Rf2* locus are significantly different in the sterile and restorer lines, which made it possible to develop KASP and CAPS markers for the analysis of individual hybrid combinations [11,15,24]. The SSR marker Xtxp50 and two intragenic CAPS markers, developed in the present study on the basis of SNPs at positions 1091 and 1092 of the Sobic.002G057050 sequence, turned out to be polymorphic between the parental lines. The A/T missense mutation at the 1090 position was found in the majority of the analyzed R-lines [11,15], which makes the marker CAPS_572 almost universal for use on a large scale. At the same time, the marker CAPS_935 is suitable only for monitoring the genetic material of the parental lines analyzed in this study because there are no data on the polymorphism of HpaI recognition sites within the Sobic.002G057050 fragment and its homologs from the pools of sterile lines, maintainer lines, and pollen fertility restorers of different origins.

The genotyping results have demonstrated the genetic diversity of the newly developed R-lines and the distinctness of them from the original restorer lines 928-1 and 929-3 (Figure 4). Since the number of sources of fertility restoration genes for breeding parental lines of hybrids is often very limited, an approach based on combinative variability has become widespread in the hybrid breeding of different crops [25,26].

Pollen fertility is a key trait in heterotic hybrid breeding. The quantity and quality of pollen produced by R-lines determines the success of seed setting during hybridization, and the yield of hybrids depends on the pollen productivity of F_1_ plants. To determine fertility, the pollen was stained and the percentage of stained pollen grains as well as their morphometric characteristics were determined using cytological preparations. In sorghum, the pollen fertility of F_1_ hybrid plants is estimated by the percentage of seeds set in isolated inflorescences. To determine viability, pollen was germinated in vitro on a culture medium [23]. The R-lines grown under controlled greenhouse conditions produced highly fertile pollen (fertility rate 97–100%). The fertility pollen indices of the earlier inbred generations of the same lines grown in the field at KES VIR, turned out to be somewhat lower and varied from 57 to 83%, which can be explained by the effects of environmental conditions (temperature, moisture availability) [27].

To develop sorghum heterotic F_1_ hybrids based on CMS, it is necessary to select parental lines with valuable traits controlled by dominant genes; for example, insect resistance. The greenbug is a key pest of sorghum, capable of destroying over 85% of the crop [28].

Several accessions resistant to *S. graminum* were found in the VIR collection. During several years of study, the resistance of these accessions was high and scored at 1–2 points [13]. In the USA, since the 90 s of the last century, the insect-resistant accessions Durra Belaya PI550610 (VIR-1362) and Dzhugara Belaya PI550607 (VIR-924) from Syria and China, respectively, which originated from VIR, have been widely involved in breeding programs [29]. Later, Wu et al. [30] analyzed the segregation of populations F_2_ and F_3_ from a cross between the Westland A line (susceptible parent) and PI550610 (resistant parent) and found two QTLs on sorghum chromosome nine (SBI-09): QSsgr-09-01 (major QTL) and QSsgr-09-02 (minor QTL), which consistently conditioned resistance to the greenbug. The authors proposed four SSR markers that could be used as markers of indirect selection. Using RNA sequencing and quantitative PCR [31], it was shown that after infestation of the resistant line PI550607 (VIR-924 according to the VIR catalogue) and the susceptible line Tx7000 with the aphid biotype Gb-I, the level of expression (overexpression) of fifteen genes in the resistant line and six genes in the susceptible one. The authors suggested that the Sobic.010G130700 gene coding RLPs (receptor-like proteins) and Sobic.003G325100 gene coding NBS protein, localized on chromosomes 10 and 3, are the most significant in the interaction of plants with aphids.

The grain sorghum accessions VIR-928 and VIR-929 from Western China have not previously been involved in breeding. As a result of the longstanding work performed using these accessions, donors of insect resistance were obtained on the base of the sterile line Nizkorosloe 81s. Lines 928-1 and 929-3 each have two highly effective dominant resistance genes. Line 928-1 has a third dominant gene, expressed against individual aphid clones [13]. There are no molecular markers yet that can effectively use the potential of these accessions.

According to our data, new sorghum lines can be recommended for producing heterotic hybrids with satisfactory grain quality. Flour from new sorghum lines in the composite mixtures has a positive effect on gluten quality and enzyme activity, so their use in breeding to produce grain sorghum hybrids for the food industry is probably promising.

## 4. Materials and Methods

### 4.1. Plant Materials, Phenotypic Analysis, and Cytological Analysis of Pollen

Eight sorghum restorer (R) lines selected from hybrids between the CMS line (A1) Nizkorosloe 81s (Ukraine) and the fertility restorer lines 928-1 and 929-3 (Dzhugara white, Western China) were studied; three lines of BC_1_F_13_ and BC_1_F_15_ (Nizkorosloe 81s × 929-3), and five lines of BC_1_–BC_2_ F_15_ (Nizkorosloe 81s × 928-1). Parental fertile lines 928-1 and 929-3 and the cytoplasmic male sterile line Nizkorosloe 81s were used as controls in estimating plant architecture-related traits and seedling–flowering period. The two sterile lines A-83 and A-10598 (India) with the A1 cytoplasmic background were also used in the experiments. At Kuban Experiment Station of VIR (KES VIR, Krasnodar Territory) in 2021 and 2022, three plants from each accession were randomly selected from the central rows, and plant height (cm) and panicle length and width (cm) were examined. To characterize the panicle shape (compact (lumpy) or spreading), an index, calculated as a ratio of the panicle length to the panicle width, was used. The period from germination to flowering was studied in 2021 at KES VIR. The data were processed statistically.

Pollen fertility of paternal plants and new lines grown in the greenhouse of the scientific and production base of the VIR Pushkin and Pavlovsk laboratories (PPL VIR, St. Petersburg, Russia, Pushkin) was assessed using a modified staining procedure with acetocarmine [32]. Cytological analysis was performed using a Zeiss Axioplan 2 Imaging microscope with an AxioVision digital camera and AxioVision 4.8 software. Pollen grains with a bright red stained cytoplasm were classified as fertile, and non-stained pollen grains were considered sterile. The percentage of fertile pollen grains was calculated based on the analysis of at least 10 fields of view at 20× magnification.

### 4.2. Experimental Plants

In the greenhouse of PPL VIR, a segregating F_2_ population was generated from the cross of the Nizkorosloe 81s line and the fertile line 929-3. In March 2021, one panicle from Nizkorosloe 81s was pollinated with the pollen of 929-3 and the F_1_ seeds were sown in March 2022. After self-pollination of the F_1_ single plant, more than 200 F_2_ seeds were obtained. Before the period of heading, the panicles of each F_2_ plant were isolated. The plants’ F_2_ were differentiated for fertile or sterile plants based on seed setting rate, and plants with a seed setting rate lower than 5% were considered sterile [7]. Flag leaf samples were taken for DNA extraction.

### 4.3. Screening for Aphid Resistance

#### 4.3.1. Adult Plant Resistance

The resistance of sorghum lines to the Krasnodar population of *S. graminum* was studied under field conditions at KES VIR in 2021 and 2022 when the most significant greenbug outbreaks were noted. The accessions (929-3, 928-1, eight new R-lines, and Nizkorosloe 81s as a susceptible control) were sown in four-meter rows. During the period of maximum pest abundance (late July–August), the damage to 20 adult plants of each line was assessed according to the following scale: 1—very slight leaf damage (<5% of the leaf surface); 3—weak (5–20%); 5—average (21–35%); 7—strong (36–50%); 9—very strong (>50%) (Figure 6) [33]. The experimental plots were not treated with insecticides.

#### 4.3.2. Seedling Resistance

The sprouted seeds were sown in rows in plastic flats filled with a non-sterile mixture of soil, sand, and peat. One tree row (20–30 plants) of the aphid-susceptible Nizkorosloe 81s line and ten rows of the tested lines were placed in each flat. The seedlings were infested with the Krasnodar aphid population, and after the death of control, plant damage was quantified using a scale: 0—no damage, 1—1–10% of the leaf surface damaged, 2—11–20% … 10—90–100%, plant death. Damage ratings 1–4 (from 1–10 to 31–40% of leaf area damaged) were referred to as resistance (R)and ratings of 9–10 as susceptibility (S) (Figure 7) [33].

### 4.4. DNA Analysis

#### 4.4.1. Microsatellite Analysis, Genotyping, and Candidate Rf Gene Identification

Genomic DNA was isolated individually from leaf tissue of 2–5 adult plants or sprouts of each line and 80 experimental plants (F_2_s) according to the protocol proposed by D.B. Dorokhov and E. Cloquet (1997) [34].

Seven polymorphic microsatellite marker loci mapped on the chromosomes 2, 3, 4, 5, 7, and 8 were used for line genotyping and analysis of segregating F_2_ population (Appendix A). SSR markers Xtxp18, Xtxp50, Xnhsbm1089, and SB2386 were used to identify alleles of the pollen fertility restoration genes *Rf1*, *Rf2*, *Rf5,* and *Rf6*, respectively. The 25 μL reaction mixture included 1.5 μL of genomic DNA (concentration approximately 200 ng/μL); 15.9 μL H2O; 2.5 μL 10X buffer; 1.5 μL 50 mM MgCl_2_; 2.4 μL 10 mM dNTP; and 1 U. Taq DNA polymerase (Dialat, Moscow, Russia), as well as 0.5 μL of 10 pM forward (F) and reverse ® primers. The forward primer was labeled with a fluorescent dye (Evrogen, Moscow). PCR was performed on an Applied Biosystems SimpliAmp amplifier (Thermo Fisher Scientific, Waltham, MA, USA) after optimizing annealing temperatures for each primer pair. PCR conditions were as follows: initial DNA denaturation at 95 °C for 3 min; further 35 cycles: 95 °C—30 s, 55 °C, or 59 °C (depending on the primer pair)—30 s and 72 °C—30 s; final elongation—5 min at 72 °C. PCR fragments with a fluorescent label were analyzed using electrophoresis under denaturing conditions on a Nanofor-05 genetic analyzer (Sintol, Moscow, Russia). The length of the fragment was determined using a molecular weight marker (S-450, Synthol). The fragments were denatured for 5 min at 95 °C before the analysis.

#### 4.4.2. CAPS Marker Development and Rf2 Alleles Confirmation

To determine the *Rf2* locus allele composition in the studied plants, we used SNPs in the translated region of the gene in the sterile line Nizkorosloe 81s and the restorer paternal lines 928-1 and 929-3. Substitutions in positions 1090 and 1091 were used to develop CAPS markers. The numbering of the sequence is according to alignment with a reference sequence Sobic.002G057050, presented in the PLAZA nucleotide sequence database. With the use of STS primers 2459403fw and 2459403rev [27], a 935 bp fragment of the *Rf2* gene localized in the range of 337–1271 bp of the reference sequence was amplified (Appendix A). PCR was carried out in a reaction mixture of 25 μL containing 50 ng of genomic DNA, 2.5 μL of 10× buffer, 1.5 μL of 50× MgCl_2_, and 0.5 μL of 10 rM primers: 2.4 μL of a mixture of nucleotides with a concentration of 2.5 mM each and 1 U Taq DNA polymerase. All reagents are from Dialat Ltd. (Moscow, Russia). PCR was carried out in a MiniAmp Plus thermal cycler (Thermo Fisher Scientific): 95 °C—3 min; 35 cycles: 95 °C—30 s, 55 °C—40 s and 72 °C—2 min; final elongation—10 min at 72 °C. SNP detection was carried out using the restriction enzymes MseI and HpaI in a volume of 10 μL following the protocol of the manufacturer SibEnzyme (Novosibirsk, Russia). Electrophoresis of restriction products was carried out in 1.5–3% agarose gel, followed by staining in ethidium bromide solution.

### 4.5. Analysis of Technological Properties of Sorghum Flour

The grains of each sorghum line grown at KES VIR in 2022 were ground in the LMT-1 mill (Moscow, Russia) and the resultant wholegrain flour was analyzed. The flours from grains of the wheat variety Pamyati Suslyakov, grown at the PPL VIR, were used as quality controls in composite samples. Protein content was determined by a quick method using a Perten Inframatic IM 9500 NIR grain analyzer in accordance with the manufacturer’s instructions. The protein complex of sorghum lines was assessed for its stable swelling ability using the sediment volume test with some modifications. In the first stage, the indices swelling in acetic acid were measured, and then, in the second stage, measurements were taken after the addition of anionic surfactant (SDS) and, ultimately, the ratio of these indicators was determined [16]. The rheological properties of sorghum–wheat composite flour were assessed with the Brabender FarinoGraph.

### 4.6. Statistical Analyses

The average value of quantitative characteristics and standard deviation were calculated using MS Excel. Pearson’s goodness-of-fit test was used for the analyzing segregation of F_2_ seed progeny.

The genetic diversity of sorghum lines was evaluated using a principal component analysis (PCoA) [35], based on the molecular binary matrix [36] in the GenAlEx 6.5 software program. SSR and CAPS markers were scored as 1 for presence, 0 for absence (Appendix A).

## 5. Conclusions

New sorghum lines isolated from hybrids between the CMS line (A1) Nizkorosloe 81s (Ukraine) and fertility restorer lines 928-1 and 929-3 (Dzhugara white, Western China) using the method of long-term saturating crossings have a number of valuable agronomic traits. Characteristics common to all eight sorghum lines studied, such as the ability to restore pollen fertility in the F_1_ generation, good pollen quality, greenbug resistance, early ripening, spreading panicle, and low stature, allow us to recommend them for producing commercial F_1_ hybrids with satisfactory grain quality for both fodder and food purposes. The lines may also be in demand in breeding programs as donors of the *Rf2* gene when breeding new restorer lines. To control seed hybridity of the F_1_ generation, to estimate line purity, and, most importantly, to analyze the population and select homozygous plants, molecular markers developed taking into account SNPs in the coding sequence of the gene or microsatellites closely linked to the gene can be used. In our study, the SSR marker Xtxp50 and two intragenic CAPS markers, developed on the basis of SNPs at positions 1091 and 1092 of the candidate gene sequence Sobic.002G057050 in the *Rf2* locus, turned out to be polymorphic between parental lines.

## Figures and Tables

**Figure 1 plants-13-00425-f001:**
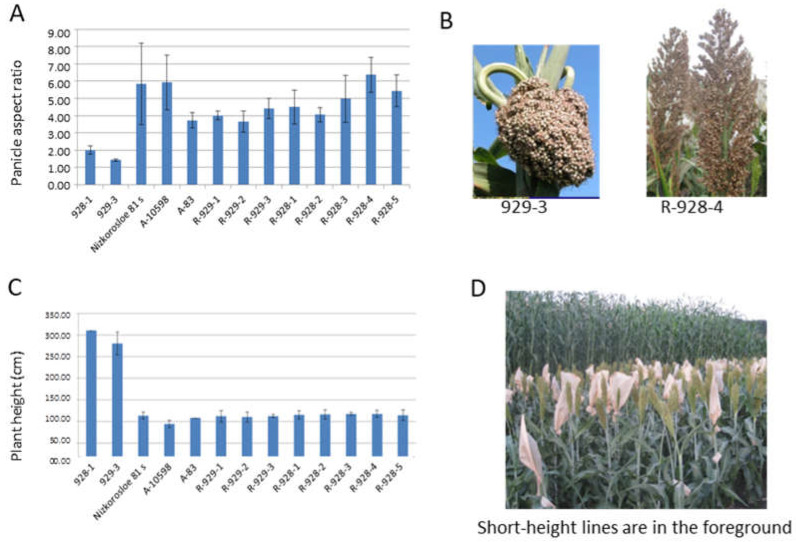
Sorghum line architecture traits: (**A**) Average mean panicle index (the ratio of the panicle length and width) and standard deviations of every sorghum line; (**B**) lumpy and loose spreading panicles of 929-3 and R-928-4 sorghum lines; (**C**) average height and standard deviation (cm) of every sorghum line; (**D**) tall and short height lines of sorghum.

**Figure 2 plants-13-00425-f002:**
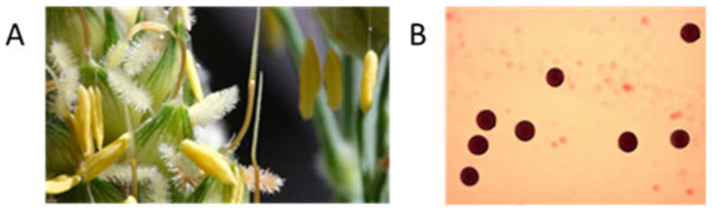
Panicle branch (**A**) and pollen grains (**B**) stained with acetocarmine at the flowering stage of the R-928-1 line plant.

**Figure 3 plants-13-00425-f003:**
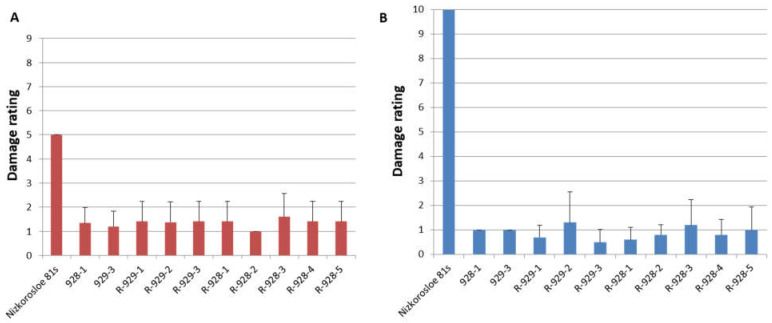
Phenotypes of sorghum lines after greenbug infestation: (**A**) Average damage score in the field in the years 2021–2023; (**B**) average damage score after infestation in the laboratory in the years 2021–2023.

**Figure 4 plants-13-00425-f004:**
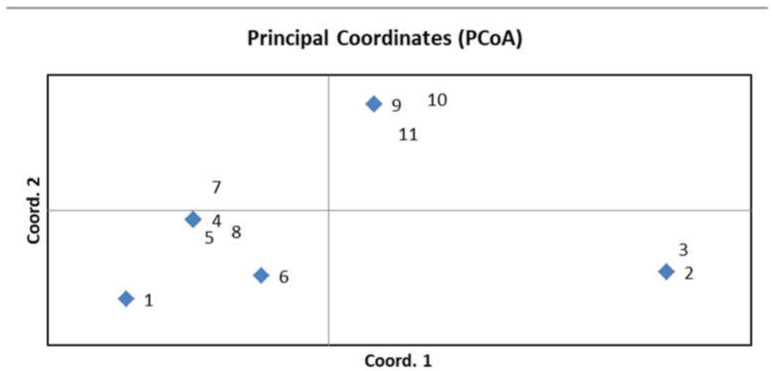
PCoA plot generated using GenALEx 6.5 showing grouping of the studied lines based on data of molecular marker analysis; 1—Nizkorosloe 81s, 2—928-1, 3—929-3, 4—R-929-1, 5—R-929-2, 6—R-929-3, 7—R-928-1, 8—R-928-2, 9—R-928-3, 10—R-928-4, 11—R-928-5.

**Figure 5 plants-13-00425-f005:**
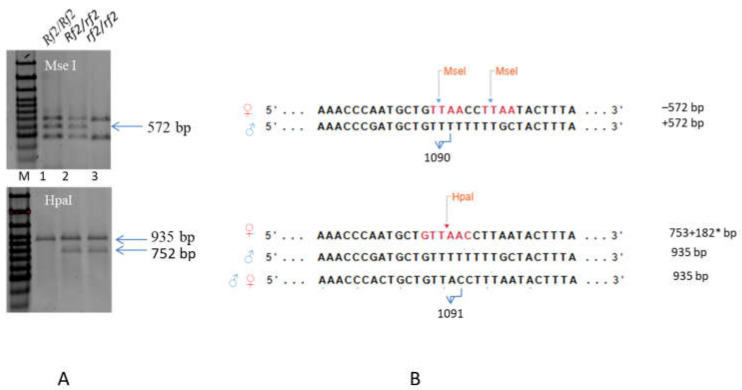
Intragenic polymorphic markers for *Rf2* gene: CAPS_572 and CAPS_935. (**A**) Polymorphism of the Sobic.002G057050 sequence fragment detected by PCR and MseI or HpaI digestion. M—Step100 Long (https://biolabmix.ru DNA marker). 1—♂ 929-3, 2—hybrid F_1_ (♀Nizkorosloe 81s × ♂ 929-3), 3—♀ Nizkorosloe 81s. (**B**) MseI and HpaI recognition sites (T↑TAA and GTT↑AAC) in Sobic.002G057050 and its homolog-aligned sequences; 572, 935, 753 + 182* bp—marker fragments in 1.5% agarose gel. * The 182 bp fragment is poorly visible in a 1.5% gel and is not shown in the figure.

**Figure 6 plants-13-00425-f006:**
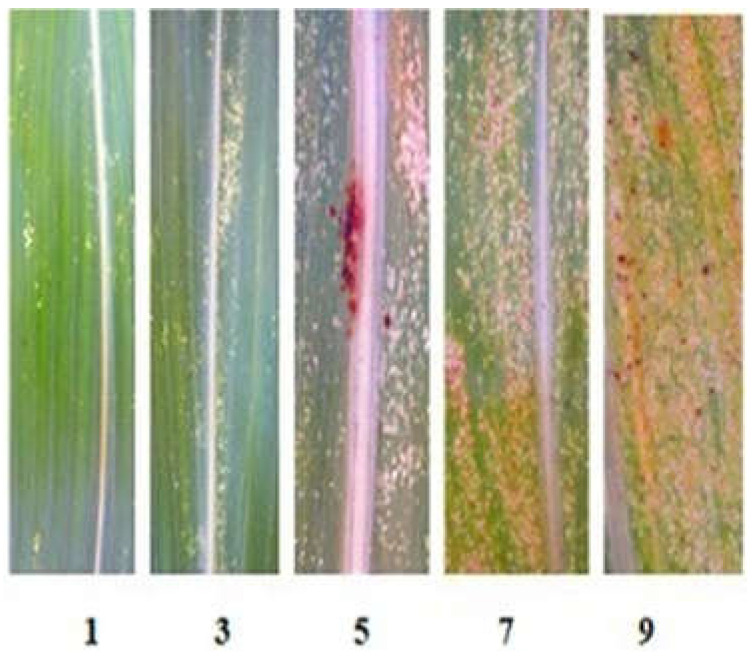
Scale (1–9) for assessing greenbug leaf damage in the field.

**Figure 7 plants-13-00425-f007:**
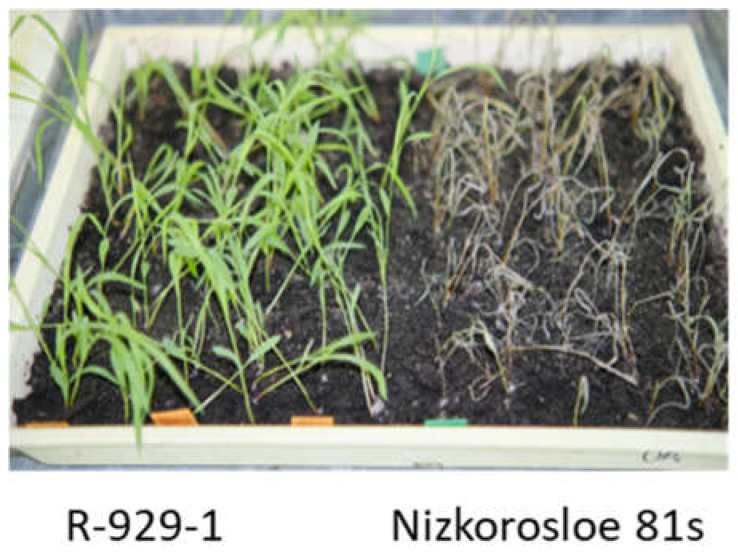
Differential responses to greenbug infestation between the resistant line R-929-1 and the susceptible line Nizkorosloe 81s.

**Table 1 plants-13-00425-t001:** Segregation by the presence (A) or absence (a) of codominant SSR markers of the genes *Rf1*, *Rf2*, *Rf5,* and *Rf6* and fertility/sterility trait in the F_2_ hybrids from crossing Nizkorosloe 81s × 929-3.

Gene, Marker (Marker Alleles)	Total Number of Plants	Phenotypic Classes of F_2_ Plants				χ^2^ for Theoretical Ratio 9:3:3:1 Data
		Aa + AA/Fert	Aa + AA/ster	aa/Fert	aa/ster	
*Rf1*, Xtxp18 (235, 237) *	80	56	12	8	4	6.30 **
*Rf2*, Xtxp50 (304, 317)	80	61	1	3	15	48.65 ***
*Rf5*, Xnhsbm1089 (227, 224)	66	41	6	16	3	5.04 **
*Rf6*, SB2386 (172, 166)	53	36	4	9	4	5.04 **

Note: Fert—fertile plant, ster—sterile plant. *d.f.* = 3, χ^2^ _0.05_ = 7.815. * The maternal A (Nizkorosloe 81s) and paternal a (929-3) alleles of the markers are indicated in the parentheses and separated by commas, ** *p* > 0.05, *** *p* < 0.01.

**Table 2 plants-13-00425-t002:** Linked inheritance of the SSR marker Xtxp50 and CAPS markers of the *Rf2* gene in the population F_2_ (Nizkorosloe 81s × 929-3).

SSR_Xtxp50 Marker	CAPS Markers: HpaI/MseI		
	935 + 753/−572 bp Homozygous for Maternal Allele	935 + 753/+572 bp Heterozygous	935/+572 bp Homozygous for Paternal Allele
304 bp *(rf2/rf2)* homozygous for maternal allele	18 plants	0	0
304 + 317 bp (*Rf2/rf2*) heterozygous	1 plant	37 plants	0
317 bp (*Rf2/Rf2*) homozygous for paternal allele	0	0	24 plants

## Data Availability

Data are contained within the article and Appendix A.

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
