# Peer review of "Newly Developed Restorer Lines of Sorghum [Sorghum bicolor (L.) Moench] Resistant to Greenbug"

_plants, 2024, doi:10.3390/plants13030425_

Round 1

Reviewer 1 Report

Comments and Suggestions for Authors

The manuscript is focused on development of restorer lines in sorghum. Restorer lines in sorghum are essential components in hybrid breeding programs, particularly for crops that exhibit cytoplasmic male sterility (CMS). CMS is a condition where the plant is unable to produce functional pollen, rendering it incapable of self-pollination. Sorghum is one of the crops where CMS is commonly exploited to develop hybrid varieties with improved traits. Overall, results are interesting and MS could be accepted for publication after a minor revision.

L36: F1, 1 should be subscript in the whole manuscript, also F2

Introduction: Why authors selected CAPS and SSR markers. There is no information about these types of markers? Need to add information in introduction section, such as  https://doi.org/10.1080/13102818.2017.1400401

Comments on the Quality of English Language

fine

Author Response

Dear Reviewer 1,

Thank you very much for taking the time to review this manuscript. Please find the detailed responses below and the corresponding revisions and corrections are indicated in red in the re-submitted files.

Comment 1:

L36: F1, 1 should be subscript in the whole manuscript, also F2

Author’s response:

It is corrected. The subscript indices were added to the F in the whole manuscript.

Comment 2:

Introduction: Why authors selected CAPS and SSR markers. There is no information about these types of markers? Need to add information in introduction section, such as https://doi.org/10.1080/13102818.2017.1400401

Author’s response:

It is corrected. The additional information on the molecular markers is added to the introduction section (new references 10, 12).

Reviewer 2 Report

Comments and Suggestions for Authors

I checked your manuscript and described comments below.

Grain sorghum is an important plant eaten all over the world.

In this paper, the author conducts phenotypic analysis and cytological analysis of pollen on Sorghum bicolor (L.) Moench. Furthermore, they performed microsatellite analysis and genotyping to identify candidate Rf genes.

These are important in sorghum research.

I think the sequence information obtained from PCR primers and DNA analysis is difficult to understand. I think it would be better to add some tables to summarize these.

I don't think this paper has new various major mistakes or grammatical problems

Author Response

Dear Reviewer 2,

Thank you very much for taking the time to review this manuscript. Please find the detailed responses below and the corresponding revisions and corrections are indicated in red in the re-submitted files.

Comment:

I think the sequence information obtained from PCR primers and DNA analysis is difficult to understand. I think it would be better to add some tables to summarize these.

We agree. We have added Table S3 “Sequence polymorphisms within the Rf2 candidate gene fragment” to the Supplementary materials. This table demonstrates SNPs in the region 1087-1097 bp of the Rf2 candidate gene fragment in the CMS- and R-lines of different origin.
